# Diagnosis of Osteoporosis by Quantifying Volumetric Bone Mineral Density of Lumbar Vertebrae Using Abdominal CT Images and Two-Compartment Model

**DOI:** 10.3390/healthcare11040556

**Published:** 2023-02-13

**Authors:** Po-Chieh Hsu, Dmytro Luzhbin, Tia-Yu Shih, Jay Wu

**Affiliations:** 1Department of Biomedical Imaging and Radiological Sciences, National Yang Ming Chiao Tung University, Taipei 112304, Taiwan; 2Institute of Statistical Science, Academia Sinica, Taipei 115201, Taiwan; 3Department of Radiology, Chung-Kang Branch, Cheng-Ching General Hospital, Taichung 407211, Taiwan; 4Department of Medical Imaging and Radiological Science, Central Taiwan University of Sciences and Technology, Taichung 406053, Taiwan

**Keywords:** osteoporosis, osteopenia, bone mineral density, opportunistic screening, two-compartment model

## Abstract

With the aging population, osteoporosis has become an important public health issue. The purpose of this study was to establish a two-compartment model (TCM) to quantify the volumetric bone mineral density (vBMD) of the lumbar spine using abdominal computed tomography (CT) images. The TCM approach uses water as the bone marrow equivalent and K_2_HPO_4_ solution as the cortical bone equivalent. A phantom study was performed to evaluate the accuracy of vBMD estimation at 100 kVp and 120 kVp. The data of 180 patients who underwent abdominal CT imaging and dual-energy X-ray absorptiometry (DXA) within one month were retrospectively collected. vBMD of L1–L4 vertebrae were calculated, and the receiver-operating characteristic curve analysis was performed to establish the diagnostic thresholds for osteoporosis and osteopenia in terms of vBMD. The average difference between the measured vBMD following TCM and the theoretical vBMD of the self-made phantom was 0.2%, and the maximum difference was 0.5%. vBMD of lumbar vertebrae obtained from TCM and aBMD obtained by DXA had a significant positive correlation (*r* = 0.655 to 0.723). The average diagnostic threshold for osteoporosis was 0.116 g/cm^3^. The sensitivity, specificity, and accuracy were 95.7%, 75.6.5%, and 80.0%, respectively. The average diagnostic threshold for osteopenia was 0.126 g/cm^3^. The sensitivity, specificity, and accuracy were 81.3%, 82.5%, and 82.7%, respectively. The aforementioned threshold values were used to perform the diagnostics on a test cohort, and the performance was equivalent to that in the experimental cohort. From the perspective of preventive medicine, opportunistic screening of bone mineral density using abdominal CT images and the TCM approach can facilitate early detection of osteoporosis and osteopenia and, with in-time treatment, slow down their progression.

## 1. Introduction

Population aging is one of the greatest current public health challenges in today’s society. After one’s prime, the individual’s bone mineral density (BMD) usually begins to decrease with an increase in age. Every 10% loss of spinal bone mineral content doubles the risk of spinal fracture [1]. Therefore, osteoporosis and osteopenia have gradually come to be recognized as important issues of international public health concern [2]. One-third of women and one-fifth of men over 50 years of age experience osteoporotic fractures, with a total of approximately 9 million people worldwide suffering from fractures owing to osteoporosis every year [3,4]. Therefore, the evaluation of bone mineral density for the early diagnosis of osteoporosis and osteopenia is crucially important as part of the preventive medicine program.

Bone density assessment methods mainly include dual-energy X-ray absorptiometry (DXA) and quantitative computed tomography (QCT). DXA is the diagnostic technology recommended by the World Health Organization (WHO). It uses the difference in the attenuation coefficients of bone minerals and soft tissues at two X-ray energies to calculate the areal BMD (aBMD) [5]. Owing to the nature of 2D projection, aBMD is affected by the patient’s body and bone thicknesses [6]. Moreover, DXA cannot evaluate the spatial distribution of bone density. On the other hand, QCT requires an equivalent bone phantom, consisting of potassium phosphate (K_2_HPO_4_) or calcium hydroxyapatite (HA), which is placed in the scanning field and used as a reference to measure patient’s volumetric BMD (vBMD). The quantitative results of QCT are more accurate and can be used to effectively evaluate the fracture risk [7,8]. However, retrospectively analyzing CT images generated for other diagnostic purposes as a means of opportunistic screening for osteoporosis is hardly feasible.

With the development of CT software and hardware, BMD analysis using conventional CT images has become feasible [9,10,11,12]. Lim et al. evaluated the severity of osteoporosis by the area proportion of a specific range of CT values in the femoral neck [10]. The analysis results were highly correlated with the T-score of DXA. Loffler et al. used the QRM phantom to establish the relationship between CT number and vBMD, showing that patients with incident vertebral fractures had lower average vBMD compared with the patients without fractures [13]. Zhu et al. performed a meta-analysis of opportunistic CT screening for osteoporosis using the measured CT number of thoracic or lumbar spines as a diagnostic threshold value [14]. The results showed that the pooled sensitivity and specificity from ten eligible studies were 0.83 and 0.74, respectively. Most of the aforementioned studies used CT numbers as a surrogate for vBMD, which could potentially lead to quantitative errors due to the energy dependence of CT numbers.

Liu et al. divided the trabecular bone into bone mineral and soft tissue using the two-compartment model (TCM) and used CT images of the femoral neck to assess bone density [15]. In this study, we used this TCM to quantitatively analyze the bone volume fraction (BVF) and vBMD of the lumbar spine. Moreover, the relationships between gender, age, and BMD were evaluated, as well as the general applicability of the TCM for the diagnosis of osteoporosis and osteopenia. The final goal was to obtain quantitative information on bone density from conventional abdominal CT images that can be used in opportunistic and preventive screening for osteoporosis.

## 2. Materials and Methods

### 2.1. Two-Compartment Model

The CT value in a CT image reflects the relationship between the linear attenuation coefficient of a mixture at a specific X-ray spectrum and the linear attenuation coefficient of water and is defined as follows:(1)$${\mathrm{CT}}_{\mathrm{mix}}=\left(\frac{{\bar{\mu }}_{\mathrm{mix}}}{{\tilde{\mu }}_{\mathrm{water}}}-1\right)\times 1000$$
where CT_mix_ denotes the CT value of the mixture, ${\bar{\mu }}_{\mathrm{mix}}$ is the average linear attenuation coefficient of the mixture, and ${\tilde{\mu }}_{w}$ is the average linear attenuation coefficient of water weighted by the X-ray spectrum. Assuming that the mixture is composed of substances a and b, the relationship between ${\bar{\mu }}_{\mathrm{mix}}$ and the linear attenuation coefficients of a and b is as follows:(2)$${\bar{\mu }}_{\mathrm{mix}}=v_{a}\times {\tilde{\mu }}_{a}+\left(1-v_{a}\right)\times {\tilde{\mu }}_{b}$$
where *v*_a_ is the volume fraction of a in the mixture, and ${\tilde{\mu }}_{a}$ and ${\tilde{\mu }}_{b}$ denote the linear attenuation coefficients of substances a and b weighted by the X-ray spectrum. Volume fraction *v*_a_ of substance a in the mixture can be expressed as follows:(3)$$v_{a}=\frac{{\bar{\mu }}_{\mathrm{mix}}-{\tilde{\mu }}_{b}}{{\tilde{\mu }}_{a}-{\tilde{\mu }}_{b}}=\frac{{\mathrm{CT}}_{\mathrm{mix}}-{\mathrm{CT}}_{b}}{{\mathrm{CT}}_{a}-{\mathrm{CT}}_{b}}$$
where CT_mix_, CT_a_, and CT_b_ are the CT values of the mixture, substance a, and substance b, respectively. Assuming that the trabecular bone is composed of cortical bone and bone marrow filling the pores [16], its BVF and vBMD can be obtained by the following equations:(4)$$B\mathrm{VF}=\frac{{\mathrm{CT}}_{\mathrm{tra}}-{\mathrm{CT}}_{\mathrm{mar}}}{{\mathrm{CT}}_{\mathrm{cor}}-{\mathrm{CT}}_{\mathrm{mar}}}$$
and
(5)$$\mathrm{vBMD}=\mathrm{BVF}\times \rho_{\mathrm{cor}}$$
where CT_tra_, CT_mar_, and CT_cor_ denote the average CT values of trabecular bone, bone marrow, and cortical bone, respectively, and *ρ*_cor_ is the physical density of cortical bone [15,17].

### 2.2. Phantom Validation for Different Tube Voltages

In our previous work [15], we measured the vBMD of a standard forearm phantom (QRM-EFP, GmbH, Moehrendorf, Germany) using the TCM method. The average differences between the TCM-based vBMD and real vBMD provided by the manufacturer at 80, 100, and 120 kVp were 0.015, 0.013, and 0.011 g/cm^3^. In this study, pure water and nine different concentrations of K_2_HPO_4_ solutions from 0.02 to 1.5 g/cm^3^ were filled into a hollow plastic phantom with five compartments to mimic the lumbar spine. Each compartment had dimensions of 3 × 1.5 × 1.5 cm. Pure water was used as the bone marrow equivalent because its CT number and physical properties are similar to those of bone marrow. The K_2_HPO_4_ solution has been frequently used in phantom calibration to correlate the CT number with vBMD. Pasoto et al. [18] measured the cortical vBMD of normal people to have an average vBMD of 0.884 and a standard deviation of 0.069 g/cm^3^. Therefore, to comprise 95% of the population (2σ), 1.0 g/cm^3^ K_2_HPO_4_ solution was used as the cortical bone equivalent. The phantom was placed at the isocenter of the CT scanner and scanned at 100 kVp and 120 kVp. In terms of image analysis, the center cross-section of the phantom was extracted, and the region of interest (ROI) covering each sector of the phantom was used to calculate the average CT value. The accuracy of the calculated vBMD and the energy dependence of the TCM method were evaluated.

### 2.3. Patient Information

This study retrospectively collected data from 180 patients who underwent abdominal CT scans and DXA examinations between December 2013 and April 2022. The intervals between CT and DXA scans were within one month. Patients who had already been diagnosed with bone metastasis, osteoarthritis, spondylitis, abnormal parathyroid function, taking drugs that affect bone metabolism, such as glucocorticoids and anticonvulsants, or receiving spinal instrumentation surgery were excluded. The patient data were randomly divided into the experimental cohort (*n* = 105) and test cohort (*n* = 75) at a ratio of 3:2. The average cohort ages were 53.2 ± 12.6 and 51.5 ± 11.7 years (*p* = 0.35), respectively. The average cohort BMIs were 23.7 ± 3.4 and 24.2 ± 4.4 (*p* = 0.42), respectively. The Ethics Review Committee approved this retrospective study (No. HP 160004), and the patients’ informed consents were exempted.

### 2.4. Analysis of DXA Data

DXA reports produced by Lunar Prodigy densitometers (GE Healthcare, Madison, WI) were collected. The aBMD measurement of the first to fourth lumbar vertebrae (L1 to L4) was performed. In terms of diagnosis, the lowest *T*-score of the four lumbar vertebrae was considered the reference standard. *T*-score ≤ −2.5 indicates osteoporosis, −2.5 < T-score ≤ −1 indicates osteopenia, and *T*-score > −1 indicates normal bone density. This classification method can increase the diagnostic sensitivity of lumbar spine bone mass measurement [19].

### 2.5. Analysis of CT Images

Abdominal CT images scanned by Aquilion 64 CT (Toshiba Medical, Tokyo, Japan) without contrast agent were retrospectively collected. The scanning parameters were 100 kVp or 120 kVp. Sagittal images were used to select the median cross-section images of L1–L4. An elliptical ROI in the central trabecular bone of the vertebral body was drawn (Figure 1) in the patient’s CT image, and the average CT value was calculated as CT_tra_ in Equation (4). The CT value of 1.0 g/cm^3^ K_2_HPO_4_ solution measured using the phantom was used as CT_cor_, whereas the CT value of pure water was used as CT_mar_. By applying Equations (4) and (5), we can calculate BVF and vBMD of vertebral trabecular bone using TCM. We further analyzed the correlation between vBMD and aBMD and performed the regression analysis to assess correlation between vBMD and gender and age.

### 2.6. Receiver-Operating Characteristic Curve

The patients in the experimental cohort were divided into three groups according to their *T*-scores: osteoporosis (*n* = 23), osteopenia (*n* = 25), and normal (*n* = 57). The receiver-operating characteristic curve (ROC) was analyzed, and the Youden index was used to evaluate the thresholds in the diagnosis of osteoporosis and osteopenia. The sensitivity, specificity, and accuracy of the proposed system were evaluated as the diagnostic performance indicators. Furthermore, the patient data in the test cohort were used to verify the performance of the proposed vBMD system. IBM SPSS Statistics 24.0 was used for statistical analysis. If the *p*-value of the two-tailed test was less than 0.05, a statistically significant difference was assumed to exist.

## 3. Results

Figure 2 shows the relationships between the theoretical vBMD of the self-made bone phantom and its CT number and between the theoretical vBMD and the measured vBMD obtained from TCM at 100 kVp and 120 kVp. At 100 kVp, the CT numbers of the phantom with different concentrations of K_2_HPO_4_ were higher than those obtained at 120 kVp. This phenomenon was significant when the theoretical vBMD was greater than 0.2 g/cm^3^, indicating that the use of the CT value for determining bone density could be affected by the tube voltage. In Figure 2(b), the average difference between the measured vBMD obtained from TCM and the theoretical vBMD was 0.2%, and the maximum difference was 0.5%. The linear fitting results at different kVp were considerably good (*R*^2^ > 0.998). No significant differences existed between the two fitting curves. Therefore, the measured vBMD obtained from TCM had no energy dependence. In addition, the slopes of the two fitting curves were close to 1, and the intercept was close to 0, indicating that the measured vBMD could deliver accurate quantitative results at 100 and 120 kVp.

Table 1 lists the basic data of the experimental and test cohorts divided into normal, osteopenia, and osteoporosis groups according to their *T*-scores. The average age, aBMD, and *T*-score in the three groups differed significantly (*p* < 0.01), whereas the BMI did not (*p* = 0.07–0.82). Figure 3 shows the fusion of CT images of lumbar vertebrae and vBMD distributions obtained from TCM of the normal, osteopenia, and osteoporosis subjects. Through ROI analysis, the average vBMD of the three vertebral trabecular bones was estimated to be 0.205 g/cm^3^, 0.137 g/cm^3^, and 0.030 g/cm^3^. Figure 4 shows aBMD and vBMD of L1–L4 in the experimental and test cohorts. aBMD and vBMD of normal subjects were greater than those in the osteopenia and osteoporosis groups. vBMD of L1–L4 in the same group did not differ significantly, whereas aBMD had a gradual upward trend. The possible cause of this could be the influence of body thickness.

Figure 5 shows the scatter plots of vBMD obtained from TCM and aBMD obtained by DXA for L1–L4 vertebrae of the experimental cohort. The Pearson correlation coefficients of vBMD and aBMD had a significant moderate positive correlation (*r =* 0.655–0.723, *p* < 0.01). It indicates that vBMD of any lumbar vertebra is suitable as an index for the diagnosis of osteoporosis. Figure 6 shows the relationship between vBMD obtained from TCM and age for men and women. vBMD shows a downward year-by-year trend with increase in age. There was no significant difference in bone mineral density between men and women in the age groups preceding the 45–54 age group. However, after that, the decline rate of women’s vBMD increased, and the average vBMD of women became significantly lower than that of men.

Figure 7 shows the results of linear regression analysis for estimating the relationship between the average aBMD and vBMD and age. aBMD of women decreased at a rate of 0.009 g/cm^2^ per year. This result was similar to that of Ardawi et al. [20]. However, aBMD of men declined considerably slower. This result might be related to the overestimation of aBMD of L3 and L4 caused by the thick waist circumference. Similarly, vBMD decreased with increase in age. The decline rate was 0.005 g/cm^3^ per year for women and 0.0031 g/cm^3^ for men. The decline rate for women was slightly higher than that for men. However, this discrepancy was not as significant as that for aBMD.

Figure 8 shows the box plots of vBMD in the normal, osteopenia, and osteoporosis groups. The average vBMD values of different lumbar vertebrae in the normal group ranged from 0.168 to 0.182 g/cm^3^; in the osteopenia group, from 0.107 to 0.124 g/cm^3^; and in the osteoporosis group, from 0.054 to 0.070 g/cm^3^. Pairwise comparison of vBMD values between different groups showed that they all were significantly different from each other (*p* < 0.01). On the contrary, vBMD values of the four vertebrae in the same group did not significantly differ from each other, indicating that any vertebra can be used to calculate vBMD.

Figure 9 shows the application of ROC analysis to evaluate the applicability of vBMD obtained from TCM as an indicator of osteoporosis and osteopenia. In the diagnosis of osteoporosis, the area under the curve (AUC) of L1–L4 vertebrae was 0.928, 0.921, 0.937, and 0.924, respectively. The average AUC of the four vertebrae was 0.930. Furthermore, the Youden index was used to evaluate the threshold in the diagnosis of osteoporosis. Its sensitivity and specificity are listed in Table 2. The L1– L4 thresholds in the diagnosis of osteoporosis were 0.121, 0.103, 0.107, and 0.114 g/cm^3^, respectively. The sensitivity was more than 87.0%, and the specificity was between 75.6% and 81.7%. In the diagnosis of osteopenia, the AUC of each vertebra was between 0.883 and 0.901, and the average AUC of the four lumbar vertebrae was 0.897. With the average threshold of 0.126 g/cm^3^, the sensitivity, specificity, and accuracy were 81.3%, 82.5%, and 82.7%, respectively.

To verify the reliability of the aforementioned diagnostic thresholds, we used the data of 75 additional patients for testing. The results are presented in Table 3. In the diagnosis of osteoporosis, the sensitivity was more than 85.7%, the specificity was between 72.1% and 83.6%, and the diagnostic accuracy was between 77.3% and 84.0%. In the diagnosis of osteopenia, the average sensitivity, specificity, and accuracy were 81.3%, 83.7%, and 82.7%, respectively. The results in the test cohort were considerably close to the diagnostic results in the experimental cohort. Therefore, using the vBMD obtained from TCM to diagnose osteoporosis and osteopenia can indeed achieve good performance.

## 4. Discussion

In this study, the measured vBMD obtained from TCM had a maximum difference of 0.5% compared with the theoretical vBMD of the self-made phantom. vBMD of lumbar vertebrae from TCM and aBMD from DXA had a moderate positive correlation. Considerably good sensitivity, specificity, and accuracy in diagnosing osteoporosis and osteopenia were demonstrated through the ROC analysis. The diagnostic thresholds were verified using a test cohort, and the performance was equivalent to that in the experimental cohort. The above findings indicate that using TCM and CT images to estimate vBMD of any lumbar vertebra can be a prospective method for opportunistic screening of bone mineral density.

In addition to high tissue contrast allowing for the detection of lesions, CT images can also be used to quantify tissue information through image analysis. Shih et al. used CT images to evaluate the radiation attenuation coefficients of different tissues [21]. Das et al. and Nakao et al. converted CT numbers into electron density for dose calculation in radiotherapy [22,23]. Many studies also confirmed that the use of CT images in the assessment of BMD has considerable potential [10,11,12,13,14]. The current clinical bone density examination procedure is mainly based on DXA, which is prone to errors introduced by the patient’s body circumference and bone thickness. In this study, we did not deliberately list vertebral fractures for exclusion, as no patients had prior reported vertebral fractures. It was, however, postfactum determined that one patient’s CT image had an L1 compressive fracture. The CT number of the vertebral body was considerably low. However, its aBMD classification was normal. This might be the reason for a decrease in the Pearson correlation coefficients of the linear fitting curve of vBMD vs. aBMD data (Figure 5). Zou et al. also concluded that aBMD was often overestimated in patients with degenerative diseases of vertebrae, resulting in false negative diagnoses [24]. Using TCM to calculate vBMD of the vertebral body allows us to avoid the influence of degenerative diseases or osteophyte formation and more accurately quantify the changes in bone mineral density.

Many studies directly analyzed the correlation between the CT number of trabecular bone and osteoporosis and deduced the diagnostic threshold of CT numbers [9,11,12,25]. However, the CT number is calculated from the attenuation coefficient of a tissue, which depends on the X-ray energy (Figure 2a). Therefore, the diagnostic criteria obtained for a specific tube voltage may not be applicable to other tube voltages or CT scanners, whereas the vBMD obtained from TCM has no energy dependence and can be applied to most operating tube voltages. Therefore, the proposed TCM method can be used in the opportunistic screening of BMD.

In this study, TCM assumes that trabecular bone is composed of cortical bone and bone marrow, and bone marrow contains soft tissue and adipose tissue. TCM can use pure water as the bone marrow equivalent because the CT value of adipose tissue is about −80 to −100, and that of soft tissue is about 50 to 70. The average CT value is close to that of water. As for the energy dependence, the linear attenuation coefficient of substances is proportional to the atomic number. The effective atomic number of water is similar to that of adipose tissue and soft tissue. The difference in energy dependence between water and bone marrow can be negligible. Therefore, this justifies using pure water as the bone marrow equivalent.

Regarding the relationship between age and bone mineral density, women’s aBMD and vBMD had a significant downward trend after age 50. The main reason is that the decrease in estrogen concentration leads to a higher rate of bone metabolism than that of bone formation [26]. This phenomenon was also observed in many DXA studies [27]. However, the trend of male aBMD decreasing with increasing age was not significant (Figure 7). This might be because men’s waists lead to the overestimation of aBMD of L3 and L4, thereby compromising the declining trend of aBMD. Therefore, certain studies focused on L1 as the diagnostic target for detecting osteoporosis [9,28]. In contrast, vBMD shows a significant inverse relationship with age and does not depend on the choice of a particular lumbar vertebra, indicating that vBMD is more suitable as an indicator of male bone mineral density.

In the ROC analysis, the sensitivity and specificity of vBMD of any lumbar vertebrae in the diagnosis of osteoporosis were no less than 87.0% and 75.6%, respectively, under the corresponding thresholds, see Table 2. The sensitivity and specificity in the diagnosing of osteopenia were no less than 81.3% and 73.7%, respectively. The evaluation performed on the test cohort also demonstrated high accuracy and reproducibility in determining the BMD. In studies with CT numbers as the diagnostic indicator [9,11], the sensitivity and specificity of diagnosing osteoporosis ranged from 75.5% to 90.3% and from 72.3% to 75.4%, respectively. The above results seem to be comparable to ours. However, their research is limited to a specific tube voltage. Therefore, using the vBMD obtained from TCM as an indicator can achieve better diagnostic performance, and the quantitative results are not affected by the tube voltage.

Nowadays, general abdominal CT scanning is one of the most commonly performed radiologic examinations. Using the TCM method to analyze existing abdominal CT images retrospectively can provide the original diagnostic information and the bone density information without additional radiation exposure to the patient. Detecting the early warning signs of osteoporosis or osteopenia is an integral part of the preventive healthcare program, as early treatment can change the trajectory of the disease and reduce medical costs. Therefore, using the TCM to evaluate bone mineral density can have great clinical value.

The main limitation of this study was that the collected data had a relatively low proportion of patients diagnosed with osteoporosis, which might have led to inaccuracy in the evaluation of diagnostic thresholds. In clinical practice, periodic equivalent phantom scanning would be necessary. The scanning frequency should be evaluated additionally according to the stability of the CT scanner and can be combined with the quarterly or annual quality assurance procedures. In the future, the patient’s own tissue can be used for TCM calibration, such that no additional equivalent substance scanning will be required.

## 5. Conclusions

The vBMD obtained from the TCM method proposed in this study quantitatively reflected the bone mineral density of the patients. Significant vBMD differences existed between groups with different degrees of osteoporosis. Any lumbar vertebra can be used to diagnose osteoporosis and osteopenia with good accuracy. Moreover, the results showed that the TCM method had high reproducibility, which was validated on a test cohort of patients. In this way, retrospective CT image analysis can be performed for opportunistic screening to improve the diagnostic value of the original CT images. As part of the preventive medicine program, it can facilitate the detection of the early signs of osteoporosis, reducing in the long run healthcare costs.

## Figures and Tables

**Figure 1 healthcare-11-00556-f001:**
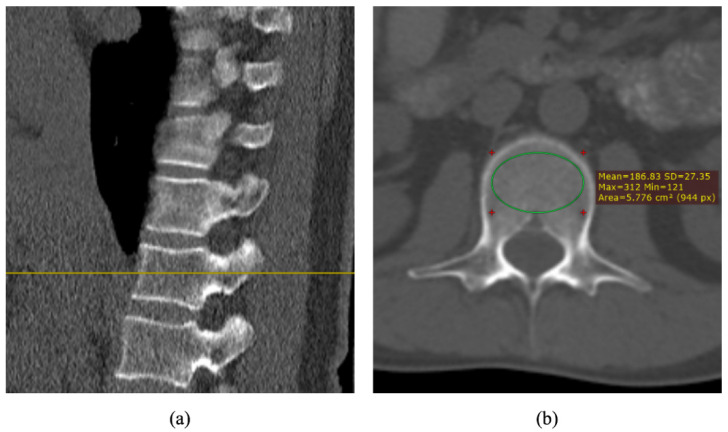
(**a**) Selection of the cross-sectional image passing through the center of the vertebral body in the sagittal plane and (**b**) circling the ROI in the cross-sectional image to include the trabecular bone.

**Figure 2 healthcare-11-00556-f002:**
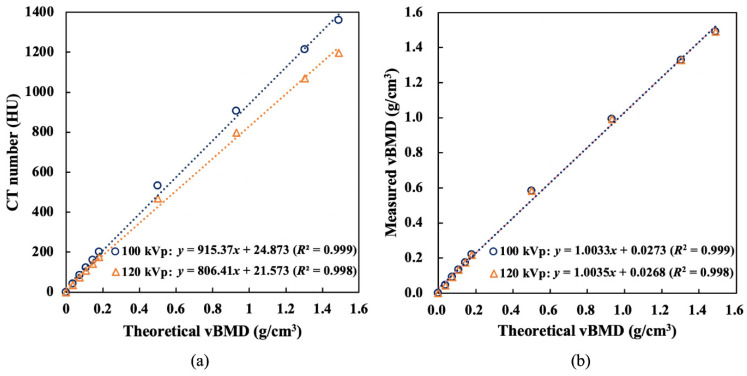
(**a**) Theoretical vBMD and CT number of the self-made bone phantom and (**b**) comparison between the theoretical vBMD and the measured vBMD obtained from TCM, scanned at 100 and 120 kVp.

**Figure 3 healthcare-11-00556-f003:**
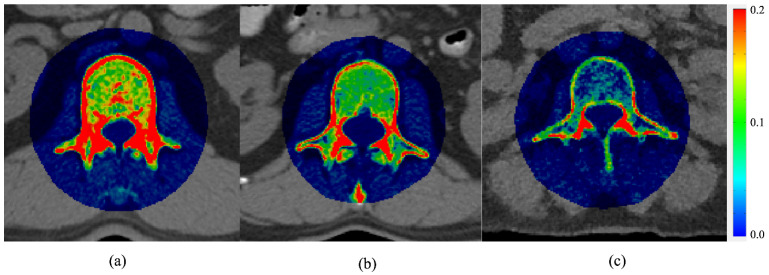
Fusion of the CT images and vBMD distributions of patients with (**a**) normal bone density, (**b**) osteopenia, and (**c**) osteoporosis. The colored area is 100% contributed by vBMD. Red and blue indicate high and low bone mineral density, respectively. The unit of color bar is g/cm^3^.

**Figure 4 healthcare-11-00556-f004:**
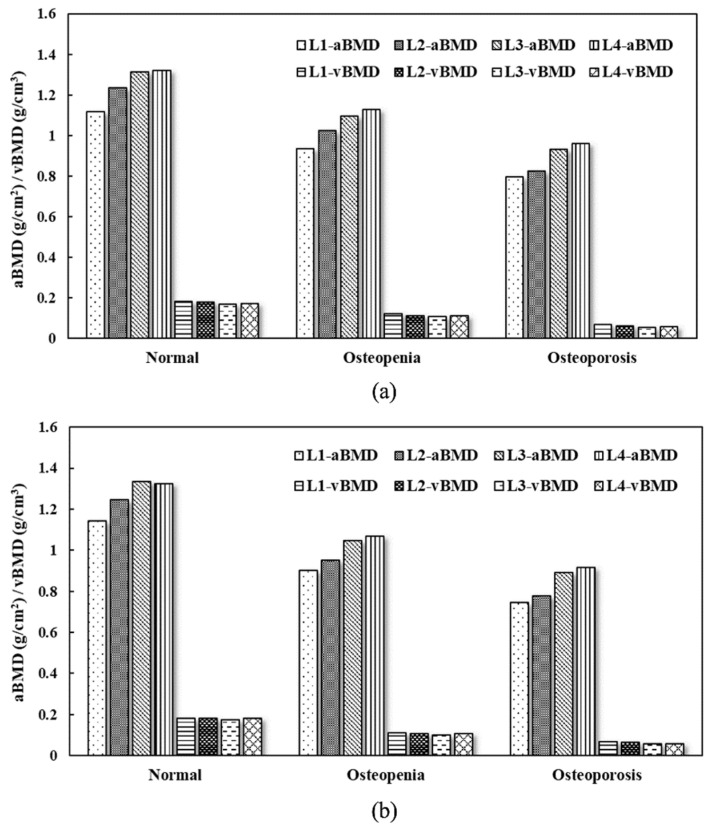
Bar graphs of aBMD (g/cm^2^) and vBMD (g/cm^3^) in the (**a**) experimental and (**b**) test cohorts. The aBMD and vBMD of the normal subjects are greater than those in the osteopenia and osteoporosis groups.

**Figure 5 healthcare-11-00556-f005:**
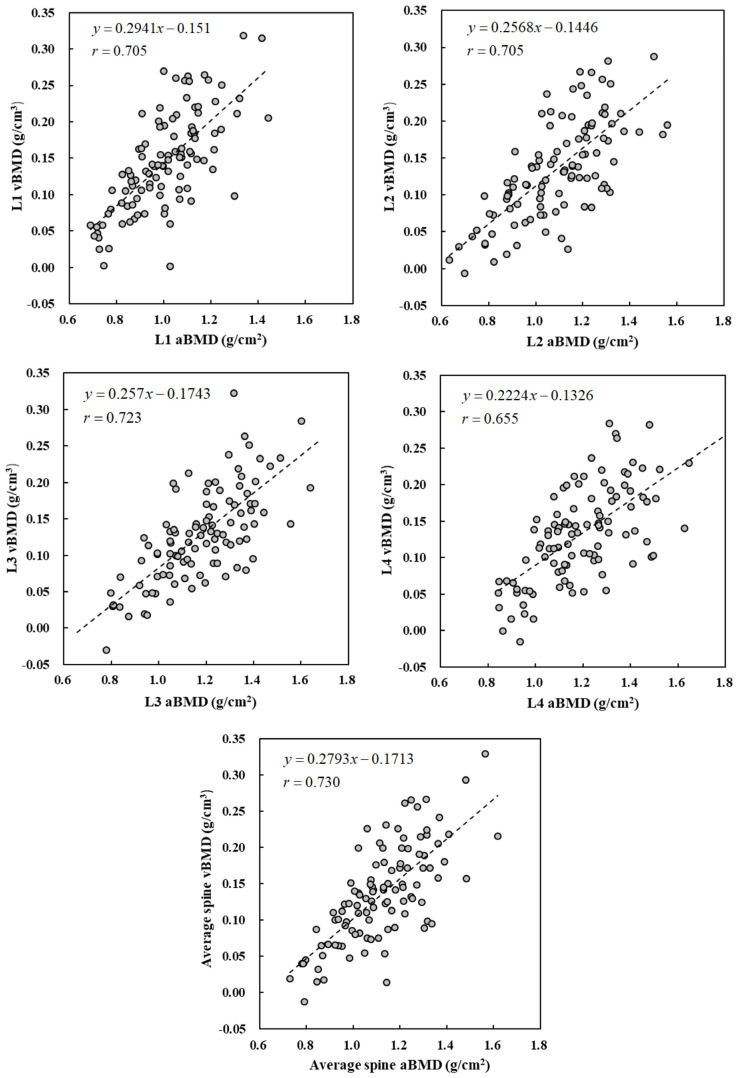
Scatter plots of vBMD and aBMD of L1 to L4 vertebrae. The Pearson correlation coefficients of vBMD and aBMD are greater than 0.655.

**Figure 6 healthcare-11-00556-f006:**
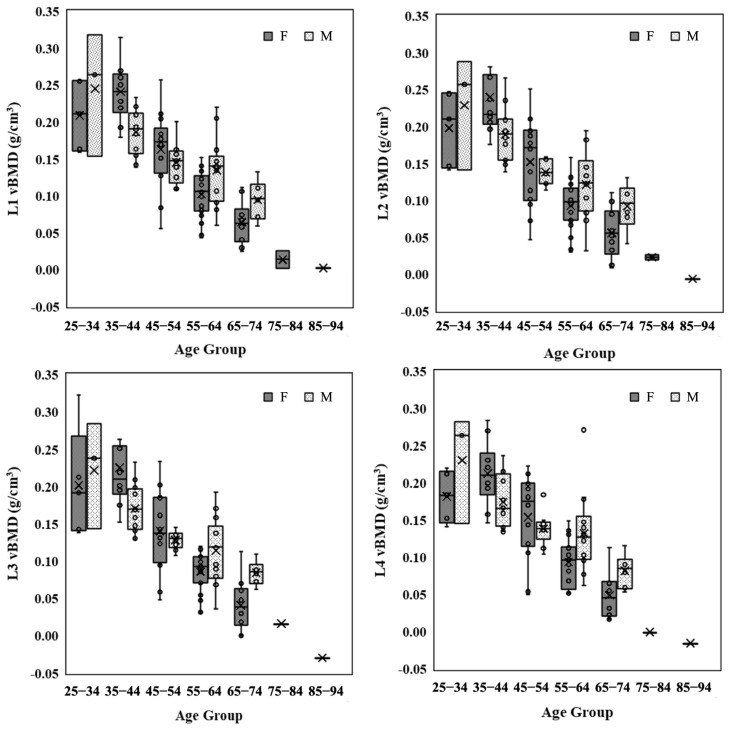
Relationships between vBMD and age group for different lumbar vertebrae. vBMD shows a downward year-by-year trend with increase in age for men (M) and women (F).

**Figure 7 healthcare-11-00556-f007:**
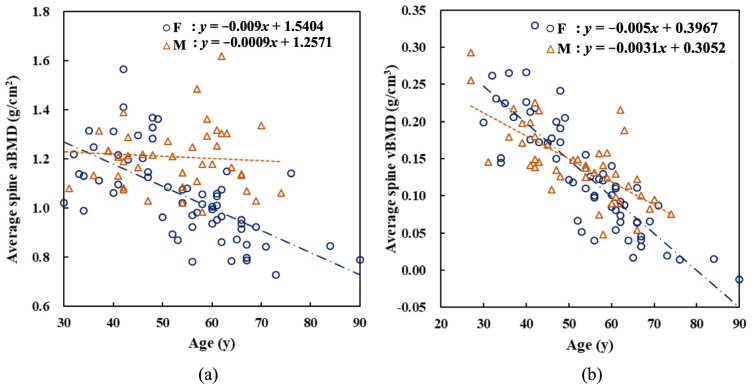
Linear fitting of (**a**) aBMD and (**b**) vBMD averaged over the four lumbar vertebrae as a function of age for men (M) and women (F). aBMD and vBMD show a downward trend with increase in age.

**Figure 8 healthcare-11-00556-f008:**
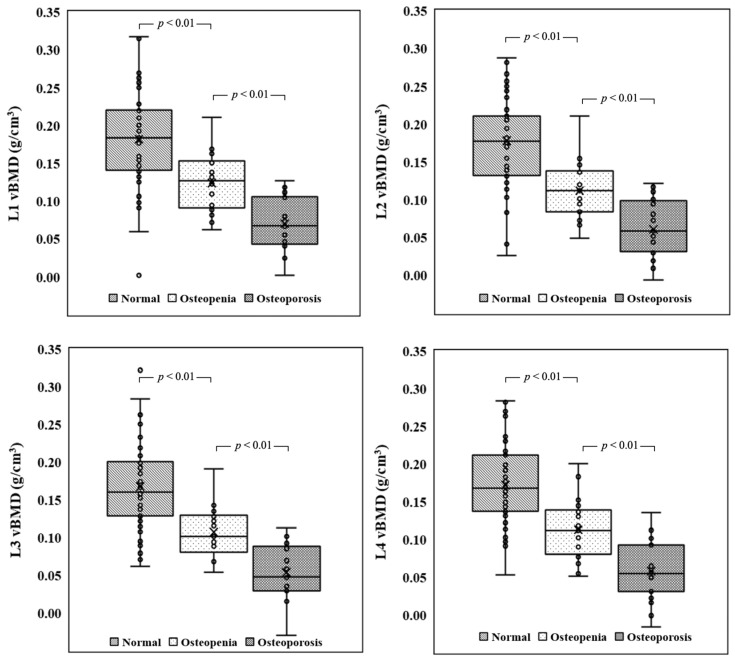
Box plots of vBMD in the normal, osteopenia, and osteoporosis groups. The upper edge of the box is the third quartile (Q3), the middle line of the box is the median (Q2), and the lower edge of the box is the first quartile (Q1). Pairwise comparison of vBMD between different groups shows that they all are significantly different (*p* < 0.01) from each other.

**Figure 9 healthcare-11-00556-f009:**
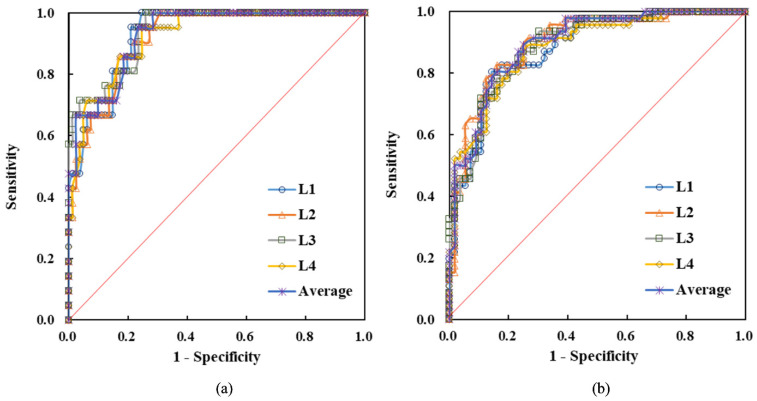
ROC analysis to estimate the diagnostic ability of vBMD in predicting (**a**) osteoporosis and (**b**) osteopenia. The average AUC for diagnosing osteoporosis and osteopenia is 0.930 and 0.897, showing considerably good results.

**Table 1 healthcare-11-00556-t001:** Basic information, aBMD, and *T*-score of three different groups in the experimental and test cohorts.

	Experimental Cohort			Test Cohort		
	Normal	Osteopenia	Osteoporosis	Normal	Osteopenia	Osteoporosis
Gender (F:M)	23:34	16:9	22:1	17:26	12:6	11:3
Age (y)	47.4 ± 11.3	55.9 ± 9.9	64.6 ± 9.1	46.6 ± 10.3	52.8 ± 10.1	64.6 ± 6.7
BMI (kg/m^2^)	23.5 ± 3.4	24.2 ± 4.3	23.7 ± 2.6	24.9 ± 4.4	23.9 ± 4.6	22.4 ± 4.2
aBMD (g/cm^2^)						
L1	1.117 ± 0.110	0.936 ± 0.069	0.796 ± 0.071	1.142 ± 0.127	0.901 ± 0.086	0.744 ± 0.114
L2	1.234 ± 0.121	1.025 ± 0.080	0.825 ± 0.086	1.247 ± 0.145	0.952 ± 0.077	0.776 ± 0.120
L3	1.314 ± 0.128	1.096 ± 0.075	0.931 ± 0.120	1.334 ± 0.159	1.048 ± 0.061	0.892 ± 0.148
L4	1.320 ± 0.153	1.127 ± 0.078	0.960 ± 0.085	1.323 ± 0.169	1.068 ± 0.096	0.917 ± 0.138
Average	1.246 ± 0.152	1.046 ± 0.105	0.879 ± 0.114	1.262 ± 0.168	0.992 ± 0.105	0.832 ± 0.147
*T*-score						
L1	0.5 ± 0.9	−1.2 ± 0.6	−2.6 ± 0.5	0.7 ± 1.1	−1.3 ± 0.7	−3.0 ± 1.0
L2	0.9 ± 1.0	−1.0 ± 0.7	−2.9 ± 0.6	1.0 ± 1.2	−1.5 ± 0.6	−3.4 ± 1.1
L3	1.4 ± 1.1	−0.5 ± 0.6	−2.1 ± 0.9	1.6 ± 1.3	−0.8 ± 0.5	−2.4 ± 1.2
L4	1.5 ± 1.2	−0.3 ± 0.7	−1.8 ± 0.7	1.6 ± 1.4	−0.6 ± 0.8	−2.2 ± 1.2
Average	1.1 ± 1.1	−0.8 ± 0.8	−2.3 ± 0.8	1.2 ± 1.3	−1.1 ± 0.7	−2.8 ± 1.2

**Table 2 healthcare-11-00556-t002:** Threshold, sensitivity, and specificity values in the diagnosis of osteoporosis and osteopenia obtained using ROC analysis with vBMD.

	Threshold (g/cm^3^)	AUC	Sensitivity (%)	Specificity (%)	Accuracy (%)
Osteoporosis					
L1	≤0.121	0.928	95.7	78.0	81.9
L2	≤0.103	0.921	87.0	81.7	82.9
L3	≤0.107	0.937	87.0	76.8	79.0
L4	≤0.114	0.924	95.7	75.6	80.0
Average	≤0.116	0.930	95.7	75.6	80.0
Osteopenia					
L1	≤0.134	0.883	81.3	84.2	82.9
L2	≤0.125	0.901	83.3	82.5	84.6
L3	≤0.129	0.889	83.3	75.4	79.8
L4	≤0.141	0.884	89.6	73.7	81.7
Average	≤0.126	0.897	81.3	82.5	82.7

**Table 3 healthcare-11-00556-t003:** Sensitivity, specificity, and accuracy of the diagnostic thresholds for osteoporosis and osteopenia calculated for the patients in the test cohort.

	Sensitivity (%)	Specificity (%)	Accuracy (%)
Osteoporosis			
L1	100.0	75.4	80.0
L2	85.7	83.6	84.0
L3	100.0	72.1	77.3
L4	85.7	75.4	77.3
Average	100.0	73.8	78.7
Osteopenia			
L1	84.4	83.7	84.0
L2	81.3	83.7	82.7
L3	81.3	79.1	80.0
L4	87.5	72.1	78.7
Average	81.3	83.7	82.7

## Data Availability

Not applicable.

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
