# Peer review of "Diagnosis of Osteoporosis by Quantifying Volumetric Bone Mineral Density of Lumbar Vertebrae Using Abdominal CT Images and Two-Compartment Model"

_healthcare, 2023, doi:10.3390/healthcare11040556_

Round 1
Reviewer 1 Report
This study attempted to establish a two-compartment model (TCM) to quantify the volumetric bone mineral density (vBMD) of lumbar spine using abdominal computed tomography (CT) images. A phantom containing water and K2HPO4 solution was used as a reference for bone mineral density in evaluating the accuracy of vBMD estimation at 100 kVp and 120 kVp. Patient study also was used to show the performance of TCM-based vBMD in diagnosing patients with osteoporosis and osteopenia confirmed by DXA.
1. In proposing the TCM-based vBMD method, a comparison study with existing technique using standard phantom is an essential necessary step.
Authors should consider an additional experiment to compare the performance of TCM-based vBMD with that of conventional technique using standard phantom.
2. This study did not give description for the configuration of phantom. The size and material composition of phantom that mimic the patient's soft tissue are known to be important to correctly calibrate the scatter and beam hardening effect and thereby to provide true reliability of vBMD values. Authors need to provide sufficiently detailed description of the phantom and to discuss the justification of the proposed phantom design.
3. The study did not describe how the calibration was made to convert the patient CT scan into vBMD. Please add detailed the calibration and patient measurement procedure.
4. The study assumed that the trabecular bone is composed of cortical bone and soft tissue equivalent of pure water with attenuation of 0 HU. However, it is known that the bone marrow in trabecular bone contains significant portion of adipose tissue, which is of much lower attenuation than 0 HU and is dependent on kV. How did authors take into account to the TCM method ?
5. Authors concluded that the TCM method was not affected by the tube voltage. This is only partly supported by study results which included only 100 and 120 kV scans. The statement should be revised.
Reviewer 2 Report
The purpose of that study was to identify osteoporosis from abdominal CT image, retrospectively collected, using a two-compartment model. The volumetric bone mineral density calculated by TCM was compared with the aBMD obtained from DXA. This is an interesting study with a new method instead of the usual CT values in Hounsfield units. The main limitations are the low sample and some methodological issues.
I have some questions and comments for the authors to consider.
L94 The sentence « Assuming that the trabecular bone is composed of cortical bone and soft tissue filling the pores » should be checked
L115 What do the authors mean by “Patients diagnosed with osteoarthritis were excluded”? Were patients with vertebral fractures also excluded?
L127 According to the WHO classification, the Tscores should be considered for the total of L1 to L4 rather than for the lowest T-score of the four lumbar vertebrae.
L156 What is the number of exams for the relationship between the theoretical and the measured vBMD obtained from TCM?
L173 As a result of misclassification, the osteoporosis group in the experimental cohort includes some patients with osteopenia and the osteopenic group includes some patients with normal BMD. The authors should rather consider the T score of the average L1L4.
The analysis of each vertebrae rather than the average of L1-L4 should be justified.
Figure 3 The level of the fusion of the CT images and vBMD distributions should be provided.
L194 Were the scatter plots and the correlation coefficients obtained from all patients? The authors should also provide those for the average of L1-L4.
L196 Considering a moderate positive association, the authors should not say “It indicates that vBMD of any lumbar vertebra is suitable to replace aBMD as an index for the diagnosis of osteoporosis ».
L234 p values should be provided
L262 The authors should begin that section with a summary of main findings
L272 Figure 8 seems not supporting “ A similar situation was also observed in this study »
Round 2
Reviewer 1 Report
My previous comments are appropriately answered. I have no additional comments.
Author Response
Dear reviewer,
We greatly appreciate your time and efforts in reviewing and helping us to improve this manuscript.
Sincerely yours,
Jay Wu
Reviewer 2 Report
The authors have satisfactorily answered to almost all questions and made the necessary changes in the manuscript. Nevertheless the second point should be corrected since the authors indicate in the discussion section one patient’s CT image showed that L1 had a compressive fracture.
Author Response
Dear reviewer,
We greatly appreciate your time and efforts in reviewing and helping us to improve this manuscript “Diagnosis of Osteoporosis by Quantifying Volumetric Bone Mineral Density of Lumbar Vertebrae Using Abdominal CT Images and Two-Compartment Model” (healthcare-2165920). Based on the valuable comments you provided, revisions have been made to address the issues. In the following, we list a point-by-point response to each of your comments.
- The authors have satisfactorily answered to almost all questions and made the necessary changes in the manuscript. Nevertheless the second point should be corrected since the authors indicate in the discussion section one patient’s CT image showed that L1 had a compressive fracture.
[Response to reviewer’s comment]:
We did not deliberately list vertebral fractures for exclusion, as no patients had prior reported vertebral fractures. It was, however, post factum determined that one patient did have a compressive fracture of L1. This special single case is discussed in the second paragraph of Discussion, and the above information is added.
Sincerely yours,
Jay Wu